# Intermediate Layers responses to Geomagnetic Activity During the 2009 Deep Solar Minimum Over the Brazilian Low Latitude Sector

Ângela M. Santos[1], Christiano G. M. Brum[2], Inez S. Batista[1], José H. A. Sobral[1], Mangalathayil A. Abdu[1], and Jonas R. Souza[1]

1National Institute for Space Research, São José dos Campos, Brazil.

2Arecibo Observatory, University of Central Florida, Arecibo, Puerto Rico.

*Correspondence to*: Ângela Santos (angelamacsantos@gmail.com; angelasantos_1@yahoo.com.br)

**Abstract.** This work presents the daytime behaviour of the Intermediate Layers (*ILs)* parameters (the virtual height - *h'IL*, and the top frequency - *ftIL)* over the low latitude region of Cachoeria Paulista (CP, 22.42° S; 45° W, I: -35.4°) during the 2009 deep solar minimim. In such a unique condition, this research reveals the *ILs'* quiet state seasonal behaviour as well as its responses to moderate changes in the geomagnetic activity. The main results show that even small variations of geomagnetic activity (quantified by the planetary Kp index) are able to modify the dynamics of the *ILs* parameters. For the first time, it was observed that during the summer, the *h'IL* decreases rapidly with the increase of geomagnetic activity, mainly in the early morning hours, while in the following hours, a smoothed rise of the *IL* was found in all seasons analysed. Regarding to the *IL* frequency, it was observed that after 12:00 LT, there is a tendency to decrease with the increase of the magnetic disturbances, being this characteristic more intense after 16:00 LT for summer and winter. For the equinox, such variation was detected, however with half of the amplitude of the other seasons. In addition, it stands out the domain of the annual periodicity of the *ftIL* while the *h'IL* presents a semiannual component under the geomagnetic quiet condition.

## 1 Introduction

The deep solar minimum of the solar cycle 23/24 provides an unprecedented opportunity to understand the variability of Earth's ambient ionosphere since 1947. During this period, an unusually

inactive state of the Sun with only relatively small sunspot-carrying active regions was observed. The solar fluxes (UV, EUV, and X-rays) responsible for the heating of the upper atmosphere and production of the ionosphere and the well-known 10.7 cm solar radio flux (F10.7cm) presented very low values when

compared to the previous solar cycle of the modern era (see for example Tapping, 2013; Tapping and Morgan, 2017; Balan et al., 2012 and Kutiev et al., 2013). During this period, reduction of ionospheric temperatures and densities were detected over several latitudes (Coley et al., 2010; Heelis et al., 2009; Yue et al., 2010; Klenzing et al., 2011; Aponte et al., 2013). The thermospheric total mass density from the prolonged minimum in solar activity between cycles 23 and 24 with that of the previous solar minima,

presented a reduction of about 10–30% compared with the climatologically expected levels (Emmert et al. 2010). Likewise, Heelis et al. (2009) and Aponte et al. (2013) reported an unprecedented contraction of the topside ionosphere to altitudes never reported before.

Great efforts have been made to better understand the behaviour of the different ionospheric layers over the equatorial and low latitudes sectors during this period. Under such condition, it is expected that

the effects caused by geomagnetic activities are highlighted, since the variability of radiation coming from the Sun in this case can be neglected. Liu et al. (2012), for example, discussed the impacts of the high-speed stream in the equatorial ionization anomaly (EIA) development. They showed that the inhibition in the EIA formation was probably due to a westward disturbance dynamo electric field. Santos et al. (2016) investigated the behaviour of the equatorial $F$ region zonal and vertical plasma drifts over Jicamarca

during the weak geomagnetic storm of June 2008. Based on a realistic low-latitude ionospheric model (SUPIM - Sheffield University Plasmasphere-Ionosphere Model), they showed that the perfect anti-correlation between the vertical and the zonal drifts close to the evening prereversal enhancement of the zonal electric field was driven mainly by a vertical Hall electric field induced by the primary zonal electric field in the presence of an enhanced nighttime E region ionization (see Abdu et al., 1998). Sreeja et al.

(2011), in their turn, showed that the daytime E-region westward drift over Trivandrum (8.5°N, 77°E; dip latitude ~0.5°N) presented a reduction that was simultaneous with the disappearance of the equatorial sporadic $E$ layer ($E_{sq}$) echoes in the ionograms. In this case, it was suggested that an additional

overshielding electric field (westward/eastward during the day/night), superposed on the ionosphere during the storm main phase, contributed to the observed reduction in the drift.

While the effects of the geomagnetic storms on the *E*, *F*, and sporadic-*E* (*Es*) layers are widely investigated, little information can be found about such effects on those layers located in the ionospheric valley, especially during the deep solar cycle minimum of 2008-2009. These layers, which are known as "intermediate layers (or just *ILs*)" are defined as a region of enhanced electron density located in the ionospheric valley that extends from the peak altitude of the daytime *E*-region to the bottom side of the

*F*-region. Fujitaka and Tohmatsu (1973) reported that the solar semi-diurnal atmospheric tide can be the dominant cause of the intermediate layers at night and that the vertical drift of the ionizations by the Sq electric field seems to modify the altitude variation of the ILs during this time. Szuszczewicz et al. (1995) found that the ILs are observed throughout the day and in all latitudes that covered the northern and southern hemispheres. Besides that, they also reported the formation of ILs at high altitudes (> 170 km)

and a monotonic descent to lower altitudes at rates as high as 8.5 km/h. Rodger et al. (1981) noted that the ILs over South Georgia (54oS, 37oW) are characterized by a prior downward movement of the F-layer, followed by the formation of the intermediate layer and its subsequent drift downwards to about 140 km. They also mentioned that initially this downward movement of the ILs can be at the same rate as the F layer, but decreases as the ILs attained lower altitudes. Mridula and Pant (2021) studied the

behavior of ILs over the equatorial location of Thiruvananthapuram and noted that the occurrence of ILs over this sector is higher in the summer and winter solstice and lower in equinoxes. They also showed that the occurrence of this layer is higher in the solar minimum than in the solar maximum period. The possible influence of the gravity waves in determining these characteristics is also discussed by the authors.

Recently, dos Santos et al. (2019) and Santos et al. (2020; 2021) have studied the essential characteristics of the *ILs* over the Brazilian sector during epochs of minimum and maximum solar activity. It was observed that these layers are predominantly diurnal and present a typical downward movement that can last from minutes to hours. Depending on the height at which the *ILs* are formed, they can descend and merge with the normal ongoing sporadic -*E* (*Es*) layers. The *IL*s' occurrence over Brazil is high and

seems to be dependent on the magnetic inclination angle and independent (or weakly dependent) on the solar activity. Nocturnal *ILs* also were observed over Brazil but they are very unusual. Regarding the shape in which the *ILs* are seen in the ionograms, it was verified that they presented a curved format similar to the "h" type *Es* layer, however the *ILs* with a straight format and spreading base appearance also were observed.

The studies conducted so far on the *ILs* over Brazil give us some indications that the dynamics of these layers can be influenced by the atmospheric tides, gravity waves, and electric fields (Nygrén et al., 1990; Wilkinson et al., 1992). The day-to-day variability in the average *ILs*' descent velocity also suggests the influence of a periodic perturbation with a periodicity of some days. The velocity values found are compatible with those of the semidiurnal and quarter-diurnal tides. However, the larger 90 descending rate (> 10 km/h) observed over the equatorial region may reveal the additional influence of the gravity waves in *IL*'s dynamics. Additionally, Santos et al. (2021) reported interesting events in which the *ILs* presented an upward movement at the same time in which the *F* layer rises due to the evening prereversal enhancement of the zonal electric field. Such characteristic was observed in most of the cases during a period of high solar activity, between October and April months, however a single case also was 95 observed in 2009. Another interesting characteristic observed is that the *ILs* could suffer in some way the influence of the prompt penetration electric fields. Dos Santos et al. (2019), for example, showed a case in which a daytime *IL* over the equatorial region of São Luis (2ºS; 44ºW) on October 9, 2009, presented a strong upward movement that carried the *IL* to the base of the *F2* layer in ~ 1.5 hours. This anomalous rise was probably caused by the joint action of the atmospheric gravity wave propagation and the dawn 100 to dusk PPEF. Santos et al. (2021) also reported the ascending *ILs*, however, during sunset times. As mentioned by the authors, it is possible that the *ILs* in these cases had been caused by the action of the PRE and in some events by the additional contribution from the prompt penetration electric fields. In all the studied events, the *ILs* were located at altitudes higher than or equal to 175 km, except the event of November 10, 2003, when an *Es* layer located at about 120 km of altitude presented an abrupt rise 105 reaching 290 km of altitude in a time interval of ~ 1.25 hours. This rapid rise of the *Es*/*IL* layers probably

was caused by an eastward electric field of ~ 0.6 mV/m arising from the PRE and the PPEF (for more details, see Santos et al., 2021).

The focus of this paper is to investigate the geomagnetic activity effects on the intermediate layers over the Brazilian low latitude sector during the deep solar minimum of 2009, regardless of the reasons why such storms were generated. As mentioned previously, this epoch is especially suited to develop studies like the one proposed here due to the very low values and little variation of the solar decimetric flux (10.7cm). In this case, the effects caused in the *ILs* by the variability of radiation coming from the Sun can be neglected, and consider only those caused by geomagnetic variations. The data and methodology used to investigate the possible influence of the geomagnetic storms in the intermediate layers is given in Section 2. The results are presented in Section 3 and finally, in Section 4, the discussion and conclusions.

## 2. Data set

In this paper, the ionospheric sounding data collected by the Digisonde operated over the low latitude site, Cachoeira Paulista (CP, 22.42° S; 45° W, I: -35.4°), during the deep solar minimum of 2009 are used to verify the possible dependence of the *ILs* on the geomagnetic activity. The ionospheric survey made by the Digisonde is based on the reflection of the electromagnetic signal transmitted vertically to the ionosphere with a peak power of the order of 10kW (for the case of Digisonde DGS256, that is the model used to collect data for 2009 over CP) at frequencies ranging from 0.5 to 30 MHz. The vertical radio sounding makes use of the fact that radio waves are reflected in the ionosphere at the height where the local cut-off frequency equals the frequency of the radio wave. The ionospheric information is recorded in the form of ionograms that display the virtual height of the returned echoes versus their frequency, generally registered at 10 and/or 15-min intervals. The Digisonde data used in this work were pre-processed through the ARTIST software (Automatic Real Time Ionogram Scaler with True Height) and also manually post-processed using the SAO-explorer software following the same criteria described by Dos Santos et al. (2019). For more details about Digisonde, see for example Reisnish (1986) and Reisnish et al. (2009). The *ILs*' virtual height (*h'IL*) and top frequency (*ftIL*) are analysed as a function

of the Kp index. All the observed *ILs* were included in the analysis, regardless if they present a descending or an ascending movement.

Before going into detail on the topic that this work proposes, we will first give an overview of the behaviour of *ILs* on the sector of CP. Figure 1 shows the variability of the parameter of frequency and height of the *ILs* (panels **a** and **b**), their distribution with the local time (panel **c**), as well as their rate of occurrence for different seasons of the year. The red, blue and grey colors are used to represent the summer (December solstice), winter (June solstice) and equinoxes, respectively. It can be observed that in general the *ILs* attain higher frequencies (> 6 MHz) after 11:00 LT (panel **a**) (except in some cases), and present a high variability in height during all the analysed period (panel **b**). The downward movement of the *ILs* is one of the important characteristic that also can be observed in panel **b**. As indicated in panel **c**, the occurrence of the *ILs* along the day is not continuous, which means that the *ILs* can appear and disappear many times during the day or simply not occur. Additionally, panel **c** also shows that in some periods (especially in winter, with some exceptions), there is a tendency that the *ILs* be formed a little later (after 08-09 LT). Regarding panel **d**, it can be seen that the *ILs* occurrence in the low latitude sector of CP increases significantly in the first hours of the day attaining its maximum at ~14:00 LT in summer, ~12:00 LT in winter, ~10:30 LT in the equinoxes. In general, the probability of occurrence decays drastically as the night-time period approaches (Note: the seasons were equally divided, i.e., 121 days around the solstices and 61 days around the equinoxes).

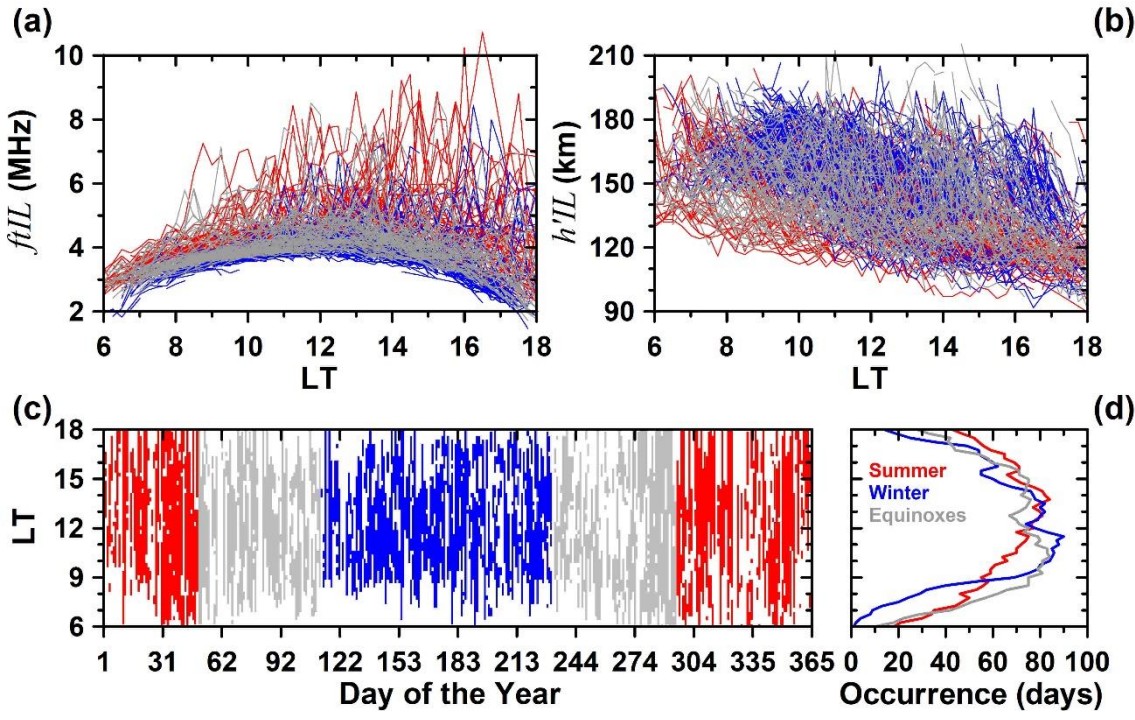

**Figure 1: Behaviour of the frequency (*ftIL*) and height (*h'IL*) parameters of the intermediate layers over Cachoeira Paulista during 2009 (panels a and b, respectively) in function of local time; distribution of the *ILs* occurrence with the local time in function of day of the year for 2009 (panel c), and; the seasonally occurrence probability of the *ILs*. In each panel, red, blue and grey are used to represent Summer, Winter and Equinoxes, respectively (panel d).**

Figure 2 summarizes the geophysical condition of the data distribution according to the solar and geomagnetic activities based on the F10.7P index and $Kp_{av}$ index, respectively. The F10.7P (grey line in top left panel) is a combination of the daily decimetric solar flux index (F10.7) and one more term (F10.7A), which corresponds to the average of the 81 previous days, thus F10.7P= (F10.7A + F10.7)/2 (given in Solar Flux Units (SFU); 1 SFU = $10^{-22}$W/($m^2$Hz)). F10.7P was chosen because several authors have shown that the ionospheric parameters are better described by this index (Brum et al., 2011 and 2012; Goncharenko et al., 2013 and references therein). In fact, Brum et al. (2011) and Brum et al. (2012) have shown that the best description of the UV-EUV (based on UV-EUV irradiance data from Pioneer Venus Orbiter (10–150 nm) and by the Solar EUV Monitor on board the Solar Heliospheric Observatory

(26–34 nm and 0.1–50 nm bands)) is given by F10.7P when compared with F10.7. In addition, their works have shown that the UV-EUV emissions tend to increase with F10.7P until a certain threshold (around 175 SFU). However, for low solar activity, the UV-EUV variations with the F10.7P can be well represented by a linear function and this feature is very important for the methodology employed in this

work, as seen below. For more details about F10.7 index, see Tapping (2013) and Tapping and Morgan (2017). The $Kp_{av}$ (grey line in left bottom panel) is the average of the 3 hours data current Kp value ($Kp_{(ref)}$) and the previous 3 and 6 hours, that is, $Kp_{av} = (Kp_{(ref)} + Kp_{(ref-3)} + Kp_{(ref-6)})/3$, which gives the standard behaviour of the geomagnetic activity and avoid sharp gradients in the temporal edges of this index (every 3 hours). Then, in the case of $Kp_{av}$, different values can be defined per day, since the *ILs* can

occur in different intervals of the day.

The occurrence number in hours of the $Kp_{av}$ level during 2009 is presented on the right bottom panel of Figure 2 (red bars). It is observed that all of the data were acquired during very low to normal geomagnetic activity ($Kp_{av} \leq 3^+$ or 3.3) according to the Wrenn et al. (1987) classification. Such distribution is very similar to that found by Terra et al. (2020) when the authors analyzed the MSTID

events for the period starting in the middle of 2018 to the end of 2019 (also low solar activity). Note that the occurrence of various levels of magnetic activity is well distributed throughout the year (left bottom panel of Figure 2) and this behaviour is the optimum condition for the kind of analysis of this work, as will be seen in this report. Tsurutani et al. (2011) have studied part of the period in analysis and showed that the causes of the low geomagnetic activity during the end of cycle #23 can be related to the solar

midlatitude small coronal holes, low IMF Bz variances, low solar wind speeds, and low solar magnetic fields. Regarding the solar activity, the period that encompasses our dataset is the end of solar cycle #23 and the beginning of solar cycle #24. A growth of activity and fluctuations of F10.7P along the year is observed, varying from 66.5 SFU to 78.1 SFU (average of 70.1 SFU, top right panel) and an uneven distribution of F10.7P (left upper panel) may be noted. Schrijver et al. (2011) showed that in agreement

with the yearly-averaged sunspot number, only 5 of 28 cycles since 1700 had a minimum lower than in early 2009. From mid-2008 until 2009/09, the fraction of spot-free days fluctuated around 82%, unprecedented in the age of modern instrumentation. Using Johann Heinrich Müller's sunspot

observations from 1709 (Figure 5 of Hayakawa et al. 2021a), and the sunspot catalog published by the Kislovodsk Mountain Astronomical Station of the Central Astronomical Observatory at Pulkovo for the recent solar cycles (1996–2019), Carrasco et al. (2021) showed that one of the most active years in the Maunder Minimum (1709), was still less active than most years in the Dalton Minimum and also less active than those of the most recent solar minima. Additionally, they mentioned that only the solar activity levels in 2008, 2009, and 2019 were similar to or lower than (as in the case of 2008) the most probable active day fraction value for 1709 (for more details see Figure 2 of Carrasco et al. 2021). This reinforces how special is of the period chosen here to analyse the possible dependence of ILs on geomagnetic activity. For more detail about Maunder Minimum, see Usoskin et al. 2015, 2021; Carrasco et al. 2021; and Hayakawa et al. 2021a,b.

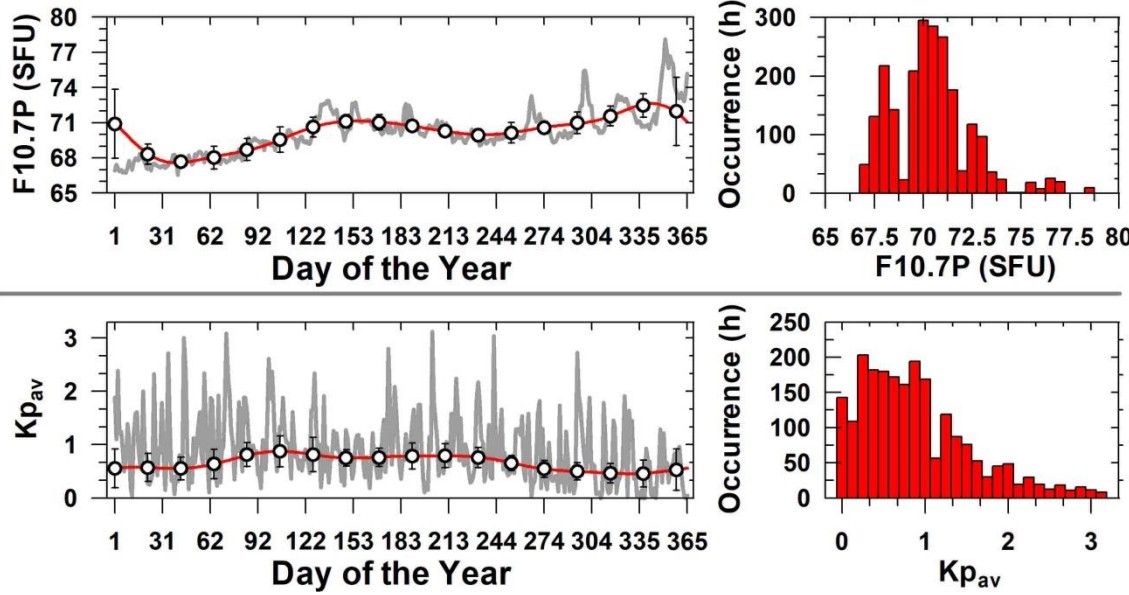

**Figure 2: Variability of the solar and geomagnetic activity quantified by the F10.7P and the Kp$_{av}$ indices (upper and bottom block of panels, respectively) for the studied period. Left panels show the geophysical conditions as a function of the day of the year while the right panels their corresponding number of occurrences (in hours) under different geophysical conditions. The dots of the left upper panel represent the 41 days' averages and the standard deviation of F10.7P, while dots of the left bottom panel represent the same range of days of the upper panel and its respective standard deviation, but for Kp$_{av}$ <=2.3 (geomagnetic condition used to construct the quiet time condition of *h'IL* and**

*ftIL*) (Note: Left panels show the geophysical conditions as a function of the day of the year (grey line)). The red continuous lines are the reconstruction of these variabilities using Fast Fourier Transform (FFT). The given occurrence in the right column of panels is the number of hours for a given interval of $Kp_{av}$ (0.125) and F10.7P (0.5 SFU).

From the *h'IL* and *ftIL* data, an empirical climatological model was developed that accounted for the dependences of these parameters on time and season, under low solar and geomagnetic activities. Determining the variability of *ILs'* parameters in function of time and season make it possible the isolation of any changes related to geomagnetic activity. The first step in our methodology was to extract the seasonal quiet time behaviour of the *h'IL* and *ftIL* parameters. For this, it was employed the weighted

arithmetic mean defined as $x(t_{ref}, d)$ represented as (equation 1):

$$\bar{x}_{(t_{ref},d_{ref})} = \frac{\sum_{d_{ref}-20}^{d_{ref}+20}\left[x_{(t_{ref},d)}(|d_{ref}-d|)\right]}{\sum_{d_{ref}-20}^{d_{ref}+20}(|d_{ref}-d|)} \tag{1}$$

where $x$ denotes *h'IL* or *ftIL* values under the geomagnetic activity condition below $Kp_{av} \leq 2.3$ for the time reference $t_{ref}$ and the selected day of the year ($d=DOY$). The average value of height and frequency of the *ILs* was calculated considering 20 days adjacent to the $d_{ref}$ and 30 minutes around the $t_{ref}$.

From the quiet time variability of the *h'IL* and *ftIL* obtained by weighted arithmetic mean process described above, a simple model was built using finite Fourier series reconstruction following the procedure by Souza et al. (2010) and Brum et al. (2011), given by

$$xV_{(t,d)} = A0_{(t)} + 2\sum_{m=1}^{4}\left[Am_{(t)}\cos(2\pi m f_1 d) + Bm_{(t)}\sin(2\pi m f_1 d)\right] \tag{2}$$

where $xV_{(t,d)}$ is the reconstructed variable as a function of time in LT *(t)* and DOY (*d*) (*xV* stands for *h'IL*

or *ftIL*), $f_1$ is the fundamental frequency of the parameter to be reconstructed (1/365), $A0_{(t)}$ is the annual average of the such parameter for a given (*t*), and finally, $Am_{(t)}$ and $Bm_{(t)}$ are the $m^{th}$ Fourier coefficients also as a function of time. The terms $A0_{(t)}$, $Am_{(t)}$ and $Bm_{(t)}$ were incorporated to the model using

polynomial fittings in function of time (LT), as shown in Figure 3, for the harmonics m=1 (one year), m=2 (~6 months), m=3 (~4 months) and m=4 (~3 months). The upper left (*h'IL*) and right (*ftIL*) panels

show the time dependence of $A0_{(t)}$ open circles for Fourier coefficients. Similarly, the values of $Am_{(t)}$ and $Bm_{(t)}$ are presented in the lower panels by the blue and red circles, respectively. In all the panels of this figure, the best polynomial fitting is represented by the continuous lines following the same colour scale described above.

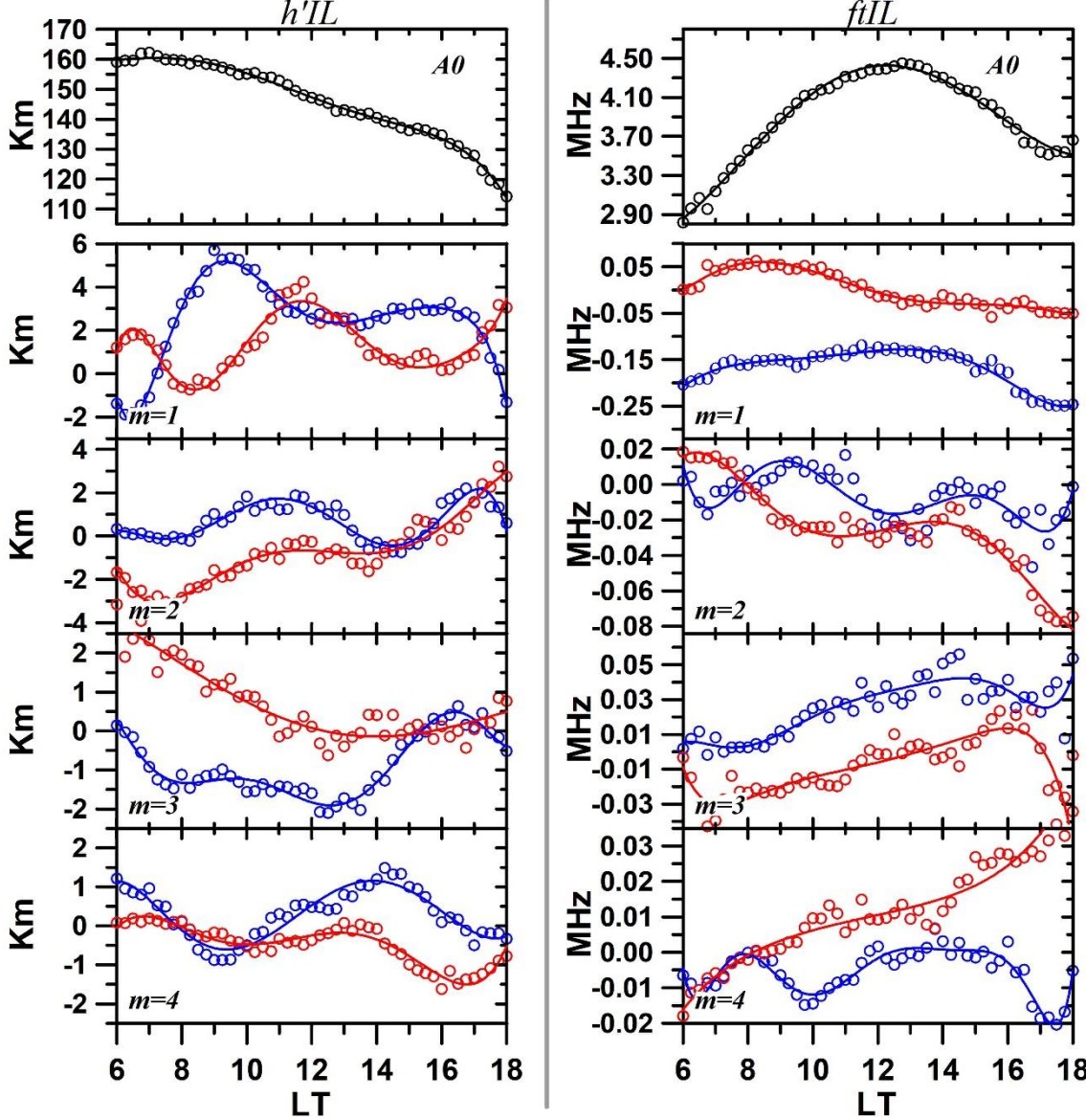


**Figure 3: Dependence of the *h'IL/ftIL*'s FFT coefficients as a function of LT (left/right, respectively). The circles are the values obtained by the FFT decomposition, while the continuous lines are the best polynomial approximation (more information in the manuscript body). The colours blue and red are used to represent the coefficients values of $Am_{(t)}$ and $Bm_{(t)}$, respectively.**

Based on the model output above described, Figure 4 shows the behaviour of the *h'IL* and *ftIL*

during the year from 06:00 to 18:00 LT (top and bottom panel, respectively). The right panels show the

dispersion diagram between the model and its respective weighted arithmetic mean (under $Kp_{av}<=2.3$)

obtained by equation 1, wherein it is possible to see the good correlation between the observation and the modeled data. The left panels show the dominance of the semi-annual and annual variation of the *ILs*' virtual height and top frequency, respectively. It is interesting to observe that the upper intermediate layers
(>160 km) are formed as winter approaches in southern hemisphere between ~ 06:00 LT and 11:00 LT, with a maximum in April/May (DOY 92-153) before the local noon. A second maximum is observed from the begging of November to middle of January (DOY 304-15), however, in a more restricted range of time (prior ~09:00 LT). After 12:00 LT, the *ILs* are generally located at altitudes at or below to 150 km. In addition, it is observed that the evolution of the *ILs* to altitudes below 120 km was more evident
between the months of April and May (DOY 92-153) at the end of the day. The bottom left panel shows an annual variation of the top frequencies, with a maximum at about 12:00-13:00 LT from November to February (DOY 304-62). It can be observed that the upper *ILs* present lower frequencies when compared to the layers located near 150 km. As the *ILs* descend, they can reach the E region and merge with the existing sporadic-*Es* layer increasing, in this way, the top frequency of the layer due to the presence of
the metallic ions.

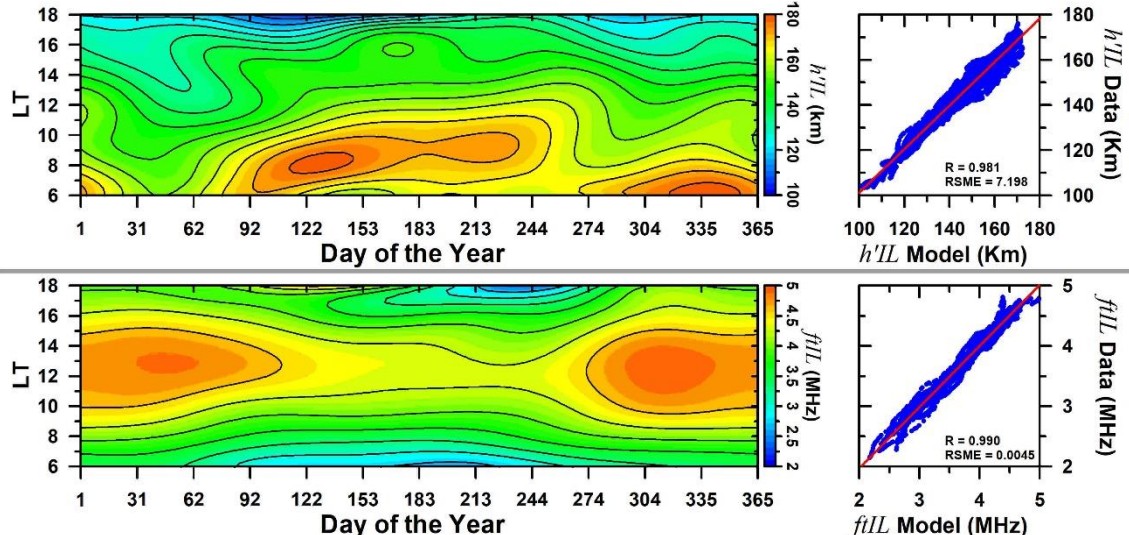

**Figure 4: Contour plot of the annual variation of the modeled virtual height (*h'IL*, left top panel) and top frequency (*ftIL*, left bottom panel) of the intermediate layers over Cachoeira Paulista. The dispersion diagrams on the right hand-side show the correlation between the weighted arithmetic mean and model results. In each panel on the right, the Correlation Coefficient (R) and Root Mean**
**Square Error (RSME) values are also provided.**

Figure 5 exemplify how the dependence of the different parameters of the intermediate layers in respect to geomagnetic activity was investigated in this work using $\Delta Kp_{av}$. The $\Delta Kp_{av}$ is the mean of the respective $Kp_{av}$ (grey line in the left bottom panel of Figure 2, $Kp_{av}= (Kp_{(ref)} + Kp_{(ref-3)} + Kp_{(ref-6)})/3$) minus the average of any value below $Kp_{av} <= 2.3$ in a range of $\pm 20$ days (this is the geomagnetic condition that the model was developed, the red line shown in the left bottom panel of Figure 2). Noticed: The usage of the residuals minimizes the background quiet time behaviour variation along the time (LT and season), enhancing this way the detection of the real contribution or not of the geomagnetic activity on the *IL*s parameters. The upper panel of Figure 5 shows the whole dataset sorted from the lowest to the highest $\Delta Kp_{av}$ values and divided into eight sections with the same percentage of samples for each range of $\Delta Kp_{av}$ (12.5%, represented by the black vertical lines) for the summer (December solstice) at 17:30 LT$\pm$30 minutes. Specifically, for this example, the selected range represents 178 data points, i.e., each 12.5% displays the behaviour of ~22 individual data samples. This panel also displays the respective F10.7P values (red line) and its respective average and standard deviation (blue open circles) for the same sorted 12.5% occurrence range of $\Delta Kp_{av}$. Note that the F10.7P mean variation for each range does not vary much which leads us to emphasize that the following variations of *ILs* are due to geomagnetic activity. The bottom panels show the *h'IL* and *ftIL* responses to the geomagnetic activity by the residual average obtained by the difference of the data and the model output presented in Figure 4 in function of $\Delta Kp_{av}$. The open circles represent the average values of the height/frequency residuals ($\Delta h'IL$ and $\Delta ftIL$, respectively) for the eight different levels of $\Delta Kp_{av}$ and their respective standard deviations (vertical and horizontal lines). The linear fitting is indicated by the blue lines. The slope (SLP) of the dependence of *h'IL* and *ftIL* with respect to the geomagnetic activity variation (km. $\Delta Kp^{-1}$ and MHz. $\Delta Kp^{-1}$) and the correlation factor (R) are also shown. In this example, it can be clearly observed that as the geomagnetic activity increases, the height of the intermediate layers also increases. The opposite occurs with the frequency when an increase of the $\Delta Kp_{av}$ causes a decrease in this parameter.

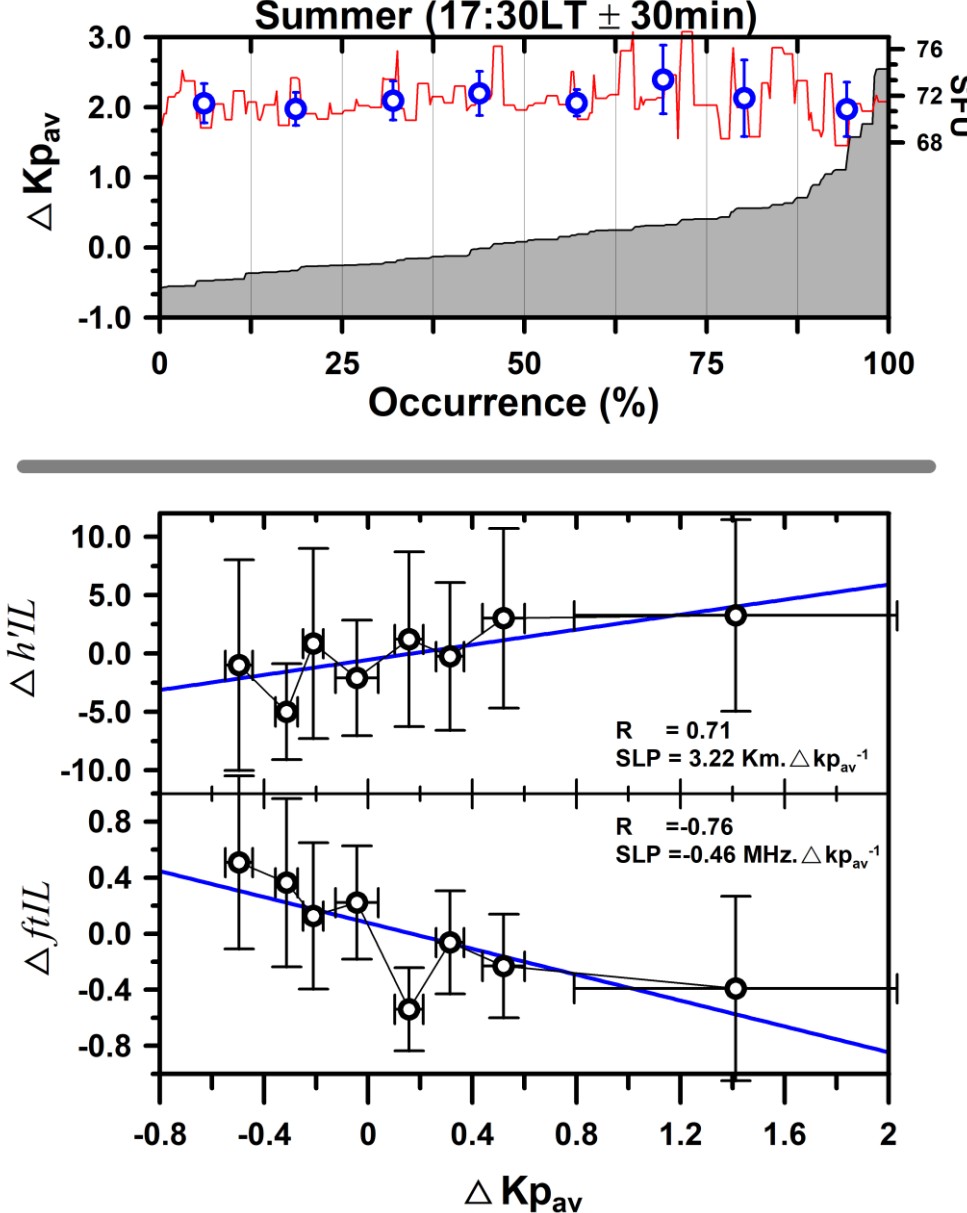

**Figure 5: Responses of the intermediate layer to the geomagnetic activity for summer at 17:30 LT±30 minutes. The upper panel shows the ΔKp_av (see more information in the text) data organized from the lowest to the highest values and divided into eight sections with the same percentage of samples. In addition, the values of F10.7P with respect to ΔKp_av and the average of the F10.7P (blue open circles) for each section are also presented. The bottom panels show the linear regression fitting over the height and frequency residual variability relative to the average ΔKp_av values.**


The same methodology explained in the case of Figure 5 was applied to all the data between 06:00 and 18:00 LT for each season. Figure 6 shows the *ILs'* dependence on the geomagnetic activity in terms of height (first two columns from left to right) and frequency (two columns of the left) for the different

seasons of the year (the data were grouped in seasons as shown in panel **c** of Figure 1). The correlation coefficient (R) of both parameters is also shown at the right column of each block of panels. The variations of the geomagnetic activity presented in this figure were 2.13±0.28 (summer), 2.18±0.24 (equinoxes) and 1.72±0.21 (winter) in Kp index. It is observed that the higher variability in *ILs'* height with geomagnetic activity occurs during the summer period. In this case, the *IL* was located lower than the expected position

with the increase of the geomagnetic activity in the beginning of the day. In the following hours, this condition decreases until a moment in which an opposite behaviour occurs, that is, a small rise of the *IL* begins to be observed with the increase of the $\Delta Kp_{av}$. Although some fluctuation in the R-value can be observed (mainly during the equinox and winter), there is a tendency that the *ILs'* height increase with the $\Delta Kp_{av}$ variation in all seasons after ~12:00 LT, as can be seen by the positive values of $km.\Delta Kp_{av}^{-1}$.

Regarding the behaviour of the frequency, it is noticed that in the summer, the *ftIL* parameter presented a tendency of increase with the geomagnetic activity in the beginning of the day (~ 0.1 $MHz.\Delta Kp_{av}^{-1}$), however from ~12:00 LT on there is a significant decrease in the *ftIL* with $\Delta Kp_{av}$ (mainly after 16:00 LT) as can be confirmed by the negative values of the coefficient correlation (R). During the equinox, the general tendency is that the *ftIL* decrease along the day, and during the winter, little or no response of the

top frequency to $\Delta Kp_{av}$ variability can be observed prior 16:00 LT and sharp decrease after the referred period similar to what is observed during the summer.

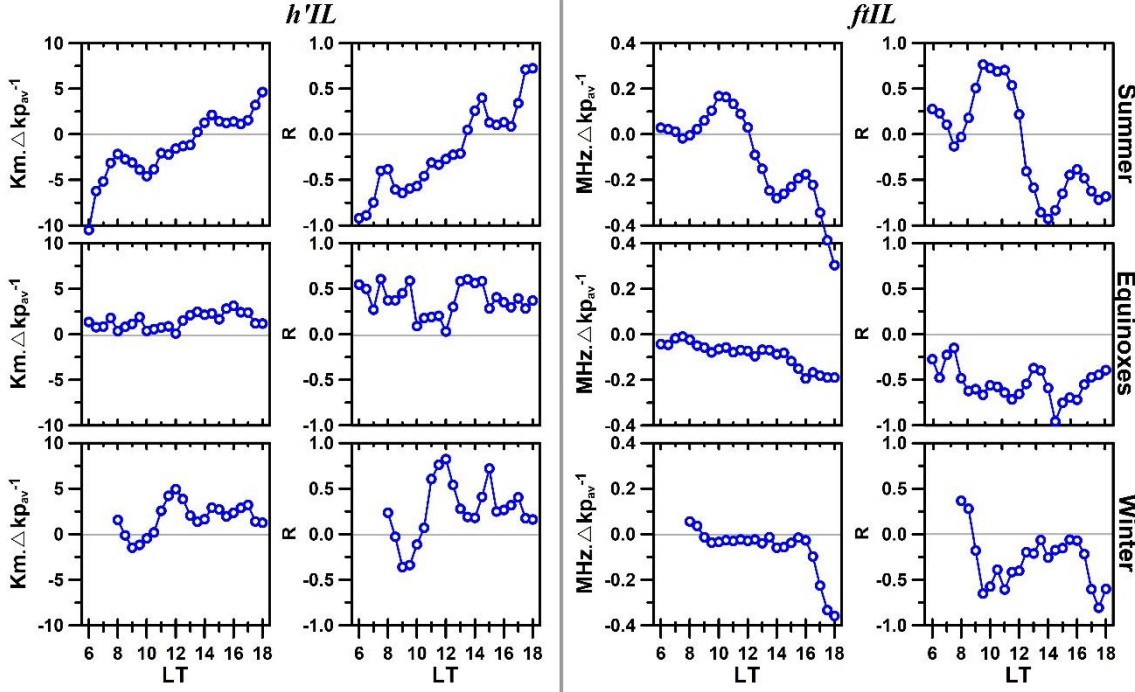

**Figure 6: Geomagnetic activity effects on *h'IL* and *ftIL* parameters for different seasons. The first two panels (from left to right) show the linear regression of the *h'IL* as a function of the $\Delta Kp_{av}$ index over the different times of the day and the correlation coefficient R. The two right panels indicate the same but for the *ftIL* parameter.**

## 4. Discussion and Conclusions.

It is well known that geomagnetic activity can drastically modify the low-latitude ionospheric dynamics. During the last solar minimum, a unique opportunity was available to investigate such dynamics, since the effects of the solar activity, that dominates the temporal variability of ionospheric properties, could be practically disregarded due to very low solar radiation variation. Using Digisonde data from a Brazilian low latitude station, Cachoeira Paulista, we studied the impacts of the geomagnetic activity in the height and top frequency of the intermediate layers during the deep solar minimum of 2009.

The results summarized in Figure 6 revealed, for the first time, that the most expressive response of the *ILs* over the low latitude region of Brazil to the geomagnetic activity occurred during the early morning hours (~06:00 - 08:00 LT) of the summer when the *ILs* presented a significant variation of their altitude with the increase of the $\Delta Kp_{av}$ (as indicated by the negative values of km.$\Delta Kp_{av}^{-1}$ in *h'IL* panels).

One of the hypotheses to explain such variation in the *h'IL* parameter is that this behaviour can be related to dusk-to-dawn directed PPEF (see for example Tsurutani et al., 2008). Such electric fields have westward polarity during daytime and therefore it may be one of the factors responsible for the occurrence of lower *h'IL* at this time. As pointed out by Santos et al. (2021), depending on the height at which the *ILs* are located, the disturbance electric field can affect considerably the vertical displacement of the intermediate layers.

Another interesting point that needs to be considered is that the movement of the *IL*s can also be influenced by the regular undisturbed day-to-day variations in the zonal electric field of the ionosphere that is directed to east/west during the daytime/nighttime hours. Therefore, it is possible that in the first 2-3 hours of our analysis period, the lowering of the *h'IL* with the increase of $\Delta Kp_{av}$ (negative values of the rate km.$\Delta Kp_{av}^{-1}$) could be a result of a competition between the eastward zonal electric field created by the E-region dynamo and the disturbance westward electric field arising from the *overshielding* process. In the following hours, an opposite situation was observed, that is, a small rise of the *ILs* occurred in all seasonal periods analyzed, as denoted by the positive values of the rate km.$\Delta Kp_{av}^{-1}$ in the first set of panels on the left in Figure 6. As mentioned by dos Santos et al. (2019), the rise of an *IL* during daytime can also be a result of the joint action of the of the eastward PPEF (*undershielding*) and gravity wave propagation. In the case studied by dos Santos et al. (2019), the rise of the *IL* also was accompanied by a decrease in their top frequency. Wakai (1967) reported that the height of the intermediate layer over Boulder (40ºN; 105ºW) also can be influenced by the magnetic disturbances, however their observation was made during the night-time.

At the same time in which a decrease in the *ILs* height is observed, an increase in the *ftIL* parameter occurs during the summer between 08:00 and 12:00 LT attaining a maximum of 0.2 MHz.$\Delta Kp_{av}^{-1}$ at 09:00 LT. It is interesting to observe that before 12:00 LT, for example, the rate variation of the *ftIL* was positive indicating that when they are located in lower altitudes, the *ILs*' top frequency increased in relation to its quiet time values. This increase in the frequency is expected since as the layer descends, it can suffer an additional increase of ionization arising from the metallic ions that contribute to the ion density in the ongoing sporadic-E (*Es*) layers. As the *ILs* presented a rise after 13:00 LT, the tendency was that the *ftIL*

decrease with the increase of $\Delta Kp_{av}$. Note that the after 16:00 LT, this decreased is more accentuated during the summer and winter. Analysing the incoherent scatter data from the mid-latitude region of Arecibo, Raizada et al. (2017) showed that the integrated electron content (*E*-region total electron content - ErTEC) between 80 and 150 km altitude regions presented a maximum variability throughout the night due to geomagnetic activity for both low and high solar activity during equinox periods. Besides that, the authors also verified that the integrated electron content during geomagnetically disturbed/normal conditions and high solar flux periods displays positive changes during summer and equinox, while it is negative in winter. Wakai (1967) reported a study about the maximum electron concentration of the nighttime *E* layer, the valley above it, and the appearance of the intermediate layer from analysis of the low-frequency ionogram obtained at Boulder on three nights of quiet, moderate, and severe geomagnetic activity. They observed an increase of the ionization in the nighttime valley at times of increased magnetic activity and the appearance of an intermediate layer in ~ 150 -160 km during periods of moderated activity. Eventually, the *IL* can be impacted by the energetic particle precipitation (EPP) (see for example Santos et al., 2016a, b), mainly during the occurrence of intense geomagnetic storms. Furthermore, as the present study refers to a period in which the geomagnetic storms were considerably weaker. That said, we believe that if *ILs* were impacted in any way by the EPP, it would not be relevant to our investigation at this moment. In addition, theoretical simulation of ion-pair production by EPP over Cachoeira Paulista have shown that the peak production of electrons is comfortable below the *IL's* minimum height used in this work (Brum et al., 2006; Brum et al., 2021).

The effects of the magnetic storms on the intermediate layer were studied also by Rodger et al. (1981) using ionosonde data from South Georgia (54ºS; 34ºW). They showed that the rate of the downward movement and the final height of the nocturnal intermediate layer are independent of the season or magnetic activity. Additionally, they observed that the probability of formation of an *IL* when the minimum virtual height of the *F2* layer is above 220 km is very low, but it can increase during magnetically disturbed periods. As shown by Santos et al. (2020), the *ILs*' occurrence over Cachoeira Paulista (22.42° S; 45° W) was very high both in 2009 (a low solar activity year and the same period of this report) and 2003 (a high solar activity year). These results show that in general, the *ILs* occurrence

resulted to be independent of the magnetic disturbances, since the referred two periods of geomagnetic activity are totally different from each other. However, their development/dynamics over Brazilian sector can be affected by disturbed electric fields, as shown by the results here presented and others previous publications (dos Santos et al. 2019, and Santos et al. 2021).

Although the impacts of the geomagnetic activity on different layers of the ionosphere have been extensively studied, there is a lack of information about what happens in the ionospheric valley region during such conditions, mainly over the low and equatorial latitudes. Using the low-power VHF radar data over the equatorial site of Jicamarca, Chau and Kudeki (2006) showed that the 150-km echoes were not affected by the electric field reversal caused by a magnetic disturbance (Kp=5). As mentioned by

them, a statistical study on the *ILs* occurrence based on the magnetic activity index Kp did not identify any correlation between magnetic activity and the 150-km echoes. On the other hand, our results show that a small variation in $\Delta Kp_{av}$ index ($\sim$ 2.0) can affect the *ILs*, especially in the morning period of the summer and late afternoon of all season over the low latitude sector over Brazil. Although the techniques used by us are different from those used by Chau and Kudeki (2006), the contrasting result reveals that

the ionospheric valley is a complex region and additional studies need to be performed to understand the physical mechanism governing the generation of the intermediate layers during the occurrence of magnetic disturbances. It is important to emphasize that for the first time it was shown that a small variation in $\Delta Kp_{av}$ index (by $\sim$ 2.0) was able to impact the dynamics of the intermediate layer over the low latitude region during the period of deep solar minimum. The main results of this work are

summarized as follow:

1. A small variation in the geomagnetic activity during low solar activity can affect both the parameter of height and frequency of the *ILs* over the low latitude Brazilian sector and such responses are dependent on local time and season;

2. During the summer, the height of the *ILs* tend to be lower with the increase of the magnetic activity

in the first hours of the day. This characteristic was probably caused by a dusk to dawn electric field;

3. During daytime, the smoothed rise of the *h'IL* can be related to the regular day-to-day undisturbed zonal electric field of the ionosphere;

4.  In respect to the top frequency dependence with geomagnetic activity, before 12:00 LT it was observed positive or null variation in all seasons. After midday, there is a tendency that the *ftIL's* decrease with the magnetic disturbances, being this characteristic more intense after 16:00 LT for the summer and winter, and;

5.  The domain of a semi-annual and an annual component variation was observed in parameters of height and top frequency of the *ILs*, respectively, for geomagnetic very quiet time condition.

*Data availability.* The Digisonde data can be downloaded in Zenodo (identified as CAJ2M 2009 in https://doi.org/10.5281/zenodo.3967542).

*Author contributions.* AMS and CGMB processed the data, performed the analysis and wrote the paper. ISB, MAA, JHAS, JRS contributed in the interpretation of the data.

*Competing interests*. The authors declare that they have no conflict of interest.

*Special issue statement.* From the Sun to the Earth's magnetosphere–ionosphere–thermosphere

**Acknowledgments**: AMS thanks the financial support from FAPESP (process number: 2015/25357-4) and CNPq (165743/2020-4). The Kp index was obtained from the World Data Center for Geomagnetism, Kyoto (http://wdc.kugi.kyoto-u.ac.jp/index.html) and Solar Radio Flux (F10.7 cm) from the National Oceanic and Atmospheric Administration (NOOA). ISB thanks CNPq grant numbers 306844/2019-2 and 405555/2018-0. One of us (JHAS) had Grant number 303383/2019-4 from the Conselho Nacional de Desenvolvimento Cientifico e Tecnológico (CNPq). J. R. Souza would like to thank the CNPq (grant 307181/2018-9) for research productivity sponsorship and the INCT GNSS-NavAer supported by CNPq (465648/2014-2), FAPESP (2017/50115-0) and CAPES (88887.137186/2017-00). The Arecibo Observatory is operated by the University of Central Florida under a cooperative agreement with the

National Science Foundation (AST-1744119) and in alliance with Yang Enterprises and Ana G. Méndez-Universidad Metropolitana.

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
