# Peer review of "Intermediate Layers responses to Geomagnetic Activity During the 2009 Deep Solar Minimum Over the Brazilian Low Latitude Sector"

_Annales Geophysicae, 2021_

## Author Response (AR1)

Anonymous Referee #1 (RC1)

We thank the reviewer for their careful reading of the manuscript and their constructive remarks. Please, see below our responses in blue.

Review of "Intermediate layers responses to geomagnetic activity during the 2009 deep solar minimum over the Brazil low latitude sector" by Santos, Brum, Batista, Sobral, Abdu and Souza

The paper has interesting results about "intermediate layers" but I find the physical explanation wanting. I suggest that the authors do a little more research to establish the physical cause of the ILs in a more convincing way.

Major Comments:

RC1-**1.** If the authors read JGR, 113, A05311, 2008, doi:10.1029/2007JA012879 carefullly, you will find that in the model/theory, northward interplanetary magnetic fields will lead to dusk to dawn electric fields that will cause downward convection of the dayside equatorial ionosphere. This is not "overshielding", but simple PPEF inputs (see their Figure 7). This electric field will cause upward convection on the nightside. Thus, if this is your scenario, the term "overshielding" is being misused. On the other hand, if the interplanetary magnetic field is southward and the dayside near equatorial ionosphere is indeed convected downward, this would indeed be due to "overshielding" effects. So in that case, the term would be used correctly.

We thank the reviewer in clarifying the definition of overshielding effects. We read attentively our manuscript and included some changes about this. For more information, please see the answer of question RC1-7.

RC1-**2.** The paper is very confusing in that both scenarios are quoted, which cannot be correct. My suggestion is to look at the interplanetary magnetic field direction during some of your IL formations and show them to your readership. Then there should not be any confusion. Unless of course you see both cases? But then you should state so in the paper.

The purpose of this work was to analyze the responses of the IL to variations of geomagnetic activity, here represented by Kp index. For this, one year of data over CP (2009) was studied in respect to Kp and some tendencies were found as the lowering or the highering h'IL with the increase of Kp when compared to the quiet day's pattern. So, in this context, we believe that it is not necessary to show separately one case or another because our focus is the statistical analysis, or in other words, what would be the more likely responses of the IL's parameters to changes in geomagnetic activity. About the confusion in the use of the term "overshielding", some changes were included in the text as mentioned above and in the answer of question RC1-7.

RC1-**3.** In any, case the above paper should be cited, which it is not at the present time.

Ok, the paper in this new version was included.

*Minor Comments:*

RC1-**4.** Lines 44 to 48. "overshielding is used in the correct sense here".

Ok.

RC1-**5.** Line 83. An upward movement of an IL would be consistent with a dawn to dusk electric PPEF caused by a southward IMF.

Ok.

RC1-**6.** Line 90-91. This is an okay description of a PPEF and southward IMF.

Ok.

RC1-**7.** Lines 103-104. Kp is not the best parameter to use to study this effect. Why don't you simple use the interplanetary magnetic field to do this study? There are many causes of geomagnetic activity, not only southward magnetic fields. Solar wind pressure pulses can cause substorms, even during northward IMFs. And some scientists believe that northward turnings of the IMF trigger substorms.

Similarly, the referee # 2 also questioned the use of the kp index instead of the Dst index. We have used the Kp index because the purpose of this paper is to investigate the responses of the IL to the overall level of geomagnetic activity, independent of how the disturbance was triggered. For all intents and purposes and to support the usage of the planetary magnetic index, we have compared the Bz direction-and-intensity and also the solar wind velocity with the kp values for the period in study (see Figure 1 below). It is clearly seen that with the increase of Bz towards the south and with the increase velocity at the same quadrant there is an increase of kp index up to 4.5, the same does not occur for Bz positive (northward), wherein the kp average is around 1.12 ±0.4 with a maximum of 1.9, which is still considered geomagnetically quiet time condition. Thus, our methodology is consistent because the periods more disturbed (for instance, over 1.12) can statistically be considered when Bz is southward and also becomes more disturbed with the increase of the solar wind velocity.

[Figure]

Figure 1. Kp average values in function of IMF's Bz component and solar wind velocity recorded by the OMNI satellite.

Similarly, to Figure 1, we have computed the dependence of Kp to variations of Dst/Sym-H values for the period in this study. Figure 2 shows the Kp average for different ranges of Sym-H (from -68nT to 16nT, steps of 2±1nT). It is clearly seen that with the decrease of Sym-H starting in 0 up to -68nT there is an increase of Kp as well. On the other hand, variations from 0 to 16nT also is noticed an increase of Kp up to Kp about 1.3, which is considered quiet condition. In a certain way, the Sym-H from 16 to -16nT nulls each other for periods of quiet condition. This statement is tested in Figure 3.

[Figure]

Figure 2. Dispersion diagram between the kp index in respect to Sym-H values for the year of 2009.

As a matter of comparison, using the same methodology proposed in the manuscript (see Figure 6 of the manuscript text), but considering now only data after 17:00 UT in order to increase the sample space (note in Figure 6 that the behavior of the IL is similar, i.e., the h'IL and ftIL tends to increase/decrease with the increase of kp, respectively) and the same range of Kp, Figure 3 below shows the variability of Sym-H, IMF Bz, solar wind velocity, ΔftIL and Δh'IL in respect to Δkp. It is clearly noticed that with the increase of ΔKp Bz increases to south while Sym-H decreases. This pattern is well

defined, evidencing that the magnetic disturbances considered in our study are associated with the direction and intensity of Bz and with the intensity of Sym-H. In this Figure, it is also seen the analysis with solar wind velocity (third panel from the top), delta frequency (fourth panel) and delta height (fifth panel). This presented analysis corroborates with the statement made by the referee and reported by JGR, 113, A05311, 2008, doi:10.1029/2007JA012879, which also corroborates with the statement of question RC1-5 *"An upward movement of an IL would be consistent with a dawn to dusk electric PPEF caused by a southward IMF"*. Summarizing, in comparison of results presented in the manuscript, we can point out that similar results were found and thus validating our study using kp index.

[Figure]

Figure 3. Variability of Sym-H, IMF Bz intensity, solar wind velocity, ΔftIL and Δh'IL in respect to Δkp.

RC1-**8.** Lines 165-166. The paper AG, 29, 839-849, 2011 should be quoted here. This paper points out the low geomagnetic activity during this extreme solar minimum, which is of importance for your paper.

The reference was included.

RC1-**9.** Line 190. See above comment.

Ok.

RC1-**10.** Lines 220-222. These altitudes are regions where precipitating electrons deposit their energy. I think you need to tell the reader why you think this is not a problem.

Figure 4 below shows the variability of cosmic noise absorption at 30MHz with respect to sunspot number obtained from 1989 to 1996 using riometer data (daytime only). Most of the absorption at this frequency occurs at D-E region altitudes over Cachoeira Paulista by non-deviative absorption with a small contribution of deviative absorption for altitudes of F-region. Note that for the low solar activity there is very little variation of the absorption, which excludes the contribution of electron precipitation under such conditions (we are excluding the proton precipitation because it occurs eastward of the South Atlantic Magnetic Anomaly, region where CP is located).

[Figure]

Figure 4. Cosmic noise absorption registered for almost one complete solar cycle over Cachoeira Paulista (from 1989 to 1996).

To confirm the statement given above, Figure 5 shows the effects of the energetic particle precipitation (EPP) (in this case electrons) in the ion-pairs formation for different fluxes and spectrums given by simulation (Brum et al., 2006; Brum et al., 2021). Clearly, the ion-pairs production is considerably higher in the D and E regions, therefore, in altitudes lower to that in which IL is commonly observed (~ 150 km). It is seen that the ion-pair formed for different ranges of energy always present an ionization peak under 100 km, even changing the flux spectrum. Thus, theoretically the electron precipitation can be neglected in our analysis discussion. However, we believe that during the occurrence of strong geomagnetic storms (that is not the case in 2009), the EPP could impact in some way the ILs. Santos et al. (2016) for example showed a case in which a layer was formed at about 150 km over Cachoeira Paulista and Fortaleza during the occurrence of magnetic storm a 23 September 2003. The appearance of this layers, that in these cases presented some degree of spread in their trace was very similar to those layers formed by the EPP (see for example Batista and Abdu (1977) and Abdu et al. (1981). So, the EPP can impact the ILs, however as we are studying the deep solar

minimum of 2009, we believe that if this influence occurred, it was not relevant. Some comments about this were include in this new version:

*"Eventually, the IL can be impacted by the energetic particle precipitation (EPP) (see for example Santos et al., 2016a, b), mainly during the occurrence of intense geomagnetic storms. Furthermore, as the present study refers to a period in which the geomagnetic storms were considerably weaker. That said, we believe that if ILs were impacted in any way by the EPP, it would not be relevant to our investigation at this moment. In addition, theoretical simulation of ion-pair production by EPP over Cachoeira Paulista have shown that the peak production of electrons is comfortable below the IL's minimum height used in this work (Brum et al., 2006; Brum et al., 2021)."*

[Figure]

Figure 5. Ion-pairs formed by electron precipitation for different fluxes and spectrums for the Cachoeira Paulista region.

**References**

Abdu, M. A.; Batista, I. S.; Piazza, l. R.; Massambani, O. *Magnetic storm associated enhanced particle precipitation in the South Atlantic Anomaly evidence from VLF phase measurements. Journal of Geophysical Research, v.86, p. 7533-7542, set. 1981.*

Brum, C.G.M. (2021), *"The impacts of particle precipitation spectrum on the 30MHz cosmic noise absorption over the under the South Atlantic Anomaly Region," 2021 XXXIVth General Assembly and Scientific Symposium of the International Union of Radio Science (URSI GASS), pp. 1-4, doi: 10.23919/URSIGASS51995.2021.9560414.*

Brum, C.G.M.; Abdu, M.A.; Batista, I.S.; Carrasco, A.J. and Santos, P.M.T. (2006). *Numerical Simulation of Nighttime Electron Precipitation in the Lower Ionosphere over a Sub-Auroral Region. Advances in Space Research, v.37, p.1051 – 1057. doi:10.1016/j.asr.2006.02.003.*

Santos, A. M., M. A. Abdu, J. R. Souza, J. H. A. Sobral, I. S. Batista, and C. M. Denardini (2016), *Storm time equatorial plasma bubble zonal drift reversal due to disturbance Hall electric field over the Brazilian region, J. Geophys. Res. Space Physics, 121, doi:10.1002/2015JA022179.*

BATISTA, I.S. AND ABDU, M.A (1977), *Magnetic storm associated delayed sporadic E*

*enhancements in the Brazilian Geomagnetic Anomaly. Journal of Geophysical Research 82: doi: 10.1029/JA082i029p04777. issn: 0148-0227.*

RC1-**11.** Lines 288-289.  Here "overshielding electric fields" and northward turnings of the IMF Bz are contradictory.

This part of the text was re-written as following:

*"One of the hypotheses to explain such variation in the h'IL parameter is that this behaviour can be related to dusk-to-dawn directed PPEF (see for example Tsurutani et al., 2008). Such electric fields have westward polarity during daytime and therefore it may be one of the factors responsible for the occurrence of lower h'IL at this time."*

RC1-**12.** Lines 303-304.  An eastward electric field would be consistent with a rise of the ILs if the IMF were southward.  However, if the IMF were northward and overshielding occurred you could get this eastward electric field as well.

This part of the text *"...depending on the height at which the ILs are located, the disturbance electric field can affect considerably the vertical displacement of the intermediate layers. They showed some cases in which the uplift of the IL was noted during sunset times considering an inversion of Bz to south (see for example their Figure 6),* therefore, in agreement with the comment of reviewer.

*Santos, A. M., Batista, I. S., Brum, C. G. M., Sobral, J.H.A., Abdu, M. A., & Souza, J. R. (2021). F region electric field effects on the intermediate layer dynamics during the evening prereversal enhancement at equatorial region over Brazil. Journal of Geophysical Research: Space Physics, 126, e2020JA028429. https://doi. org/10.1029/2020JA028429.*

RC1-**13.** Lines 310-313.  Such shielding/overshieding competition has not been observed in major magnetic storms caused by sheaths and ICMEs.  See example in GRL, 32, L12S02, 2005. Doi:10.1029/2004GL021467.  On the other hand, if these events occurred in high speed solar wind streams, such IMF north-south reversals are common.  See JGR, 111, A07S01, 2006.  Doi:10.1029/2005JA011273.   Both of these examples are typical. It would help if you identified what type of solar winds your geomagnetic activity occurred in.

As shown in Figures 1 and 3, the highest geomagnetic condition occurred when Bz was to the south and increased with the increase of Bz intensity too. A point that may have been overlooked by the reviewer is that we are not using the actual kp values, but an average of the actual and the 6 hours before, which implies that the geomagnetic conditions represented here are very stable in time and IMF reversals will not greatly impact in our final results. Again, we don't want to know the origins of the geomagnetic conditions, but the statistical responses of the IL's to that.
* * *
We would like to thank the referee for his/her contributions to the improvement of our paper. We tried to respond to every question and we made various changes according to some of the reviewer requests. Corrections and enhancements appear in blue on the text.

Referee Report for the AnGeo-2021-52,

Summary:

This article interestingly describes a case report for the intermediate layers to the geomagnetic activity over the Brazilian sector during the deep solar minimum of SCs 23/24 (2009). While the authors show unique data and discussions, their descriptions look slightly excessive and require additional justifications and modifications.

Major Comments:

**RC2-1.** The authors described their target interval (2009) as "the deepest solar minimum of the last 500 years" (e.g., P1L13-14). This is not true. Recent studies have proven that the solar activity was much more quiet during the Maunder Minimum than during 2009, on the basis of the cosmogenic isotopes (DOI: 10.1051/0004-6361/201526652; DOI: 10.1051/0004-6361/202140711), the sunspot records (DOI: 10.1093/mnras/stab1155; DOI: 10.3847/1538-4357/abd949), and the visual coronal structures (DOI: 10.1051/swsc/2020035). The authors should explicitly compare this deep minimum with the Maunder Minimum, to better contextualize their result in the longer-term space climate studies.

Some comparisons between this deep minimum with Maunder Minimum were included in the text as can be seen below:

*"Schrijver et al. (2011) showed that in agreement with the yearly-averaged sunspot number, only 5 of 28 cycles since 1700 had a minimum lower than in early 2009. From mid-2008 until 2009/09, the fraction of spot-free days fluctuated around 82%, unprecedented in the age of modern instrumentation. Using Johann Heinrich Müller's sunspot observations from 1709, Carrasco et al. (2021) concluded that one of the most active years in the Maunder Minimum (1709), it was still less active than most years in the Dalton Minimum and those of the most recent solar cycles. As commented by the authors, the solar activity level in 2009 was similar to that in 1709 according to its most probable value. All the characteristics mentioned above reinforce the importance of the period chosen here to analyze the possible dependence of ILs on geomagnetic activity since 2009 can be in some way comparable to the Maunder Minimum epoch of greatly suppressed solar activity and considered as the weakest cycle of the past 100+ years (Zharkova, 2020)".*

**RC2-2.** In this context, the authors should also address why the solar minimum in 2008/2009 was that significant. The authors have cited F10.7, whereas this lasted only after 1947 according to Ken Tapping's works (DOI: 10.1002/swe.20064; DOI: 10.1007/s11207-017-1111-6), which is missing in their reference list. This minimum was somewhat comparable with the deep minima of SCs 24/25 and SCs 13/14 (DOI: 10.1007/s11207-016-1014-y; DOI: 10.1093/mnras/stab1155). The authors should explicitly address such a long-term context.

We thank reviewer by these comments. Please see our answer in the previous question (RC2-1). The Tapping 's works are now cited in our manuscript as suggested by the reviewer.

**RC2-3**. The authors have used 3 paragraphs of their introduction to (mainly) describe their own studies. The readership would wish to know if this topic is only researched in their laboratory. Therefore, I have to strongly recommend the authors to address other teams' achievements. Otherwise, the authors have to explicitly clarify why other teams have not researched this topic.

The reviewer has reason when says that there are at least 3 paragraphs describing our own results. These paragraphs were included in the manuscript because we thought it is important to present a review for the readers about the behaviour of ILs over Brazil as it was mentioned in the first version of the manuscript (page 3, lines 55-56 in this new version). This is because there is a very limited number of studies showing the behavior of ILs using Digisonde data. We were the first authors to give focus to this specific region over Brazil using this kind of instrument. In the discussion section, the reviewer can find the citation of other papers of different authors.

RC2-4. The authors should describe more about the digisond dataset. From when to when these data are available? Which instruments were used here? How these data have been calibrated? It would be better to let the readership to understand the data within this manuscript.

The Digisonde (Digital Ionospheric Goniometric IonoSONDE) is an ionospheric radar that uses high-frequency radio waves for the remote sensing of the ionosphere. It is composed by a transreceiver system that emits pulses of electromagnetic energy at frequencies ranging from 0.5 to 30 MHz. As shown in Figure 1, simultaneous ionospheric observations are made at each 5 -15 minutes in more than 60 locations around the globe.

[Figure]

Figure 1 – Distribution of Digisondes installed around the world.

Although this instrument does not provide the electron density profile in the valley region where the ILs usually occur, some interesting characteristics of the general behavior of the ILs can be explored. Essentially, the ionospheric survey made by this instrument is based on the reflection of the electromagnetic signal transmitted vertically to the ionosphere with a peak power of the order of 10kW (for the case of Digisonde DGS256, that is the model used to collect data from 2009 over CP) at frequencies ranging from 0.5 to 30 MHz. The vertical radio sounding makes use of the fact that radio waves are reflected in the ionosphere at the height where the local cutoff frequency equals the frequency of the radio wave. The ionospheric information is recorded in the form of ionograms that display the virtual height of the returned echoes versus their frequency, registered at 10 and/or 15-min intervals. The Digisonde data used in this work were preprocessed through the ARTIST software (Automatic Real Time Ionogram Scaler with True Height) and after manually scaled by the SAO-explorer software using the same criteria described by Dos Santos et al. (2019). For more details about Digisonde, see for example Reisnish (1986) and Reinish et al. (2009). All the aforementioned description of the Digisonde and data reduction method is very known by the community, because of this we neglected further information about this in our manuscript. About the description of Digisonde, see also the site *https://ulcar.uml.edu/digisonde_dps.html* and the references cited there. We included a briefly description about the Digisonde as can be see below:

*"The ionospheric survey made by the Digisonde is based on the reflection of the electromagnetic signal transmitted vertically to the ionosphere with a peak power of the order of 10kW (for the case of Digisonde DGS256, that is the model used to collected data from 2009 over CP) at frequencies ranging from 0.5 to 30 MHz. The vertical radio sounding makes use of the fact that radio waves are reflected in the ionosphere at the height where the local cut-off frequency equals the frequency of the radio wave. The ionospheric information is recorded in the form of ionograms that display the virtual height of the returned echoes versus their frequency, generally registered at 10 and/or 15-min intervals. The Digisonde data used in this work were pre-processed through the ARTIST software (Automatic Real Time Ionogram Scaler with True Height) and after manually scaled by the SAO-explorer software using the same criteria described by Dos Santos et al. (2019). For more details about Digisonde, see for example Reisnish (1986) and Reisnish et al. (2009)."*

**References**

Reinisch, B. W.: New techniques in ground-based ionospheric sounding and studies, Radio Sci., 21, 331–341, 1986.

Reinisch, B. W., Galkin, I. A., Khmyrov, G. M., Kozlov, A. V., Bibl, K., Lisysyan, I. A., ... & Luo, Y. New Digisonde for research and monitoring applications. Radio Science, 44(01), 1-15, 2009.

RC2-5. Why do the authors use the Kp index here? The authors should redo their analyses, replacing the Kp index with the Dst index (or at least some more quantitative indices). Here, they have to explain why the authors chose the specific geomagnetic index.

Referee #1 also questioned about not using a different proxy for our analysis as well. In his/her case was suggested to use IMF parameters instead kp. We have used the Kp index because the purpose of this paper is to investigate the responses of the IL to the overall level of geomagnetic activity, independent of how the disturbance was triggered. For all intents and purposes and to support the usage of the planetary magnetic index, we have compared the Bz direction-and-intensity and also the solar wind velocity with the

kp values for the period in study (see Figure 1 below). It is clearly seen that with the increase of Bz towards the south and with the increase velocity at the same quadrant there is an increase of kp index up to 4.5, the same does not occur for Bz positive (northward), wherein the kp average is around 1.12 ±0.4 with a maximum of 1.9, which is still considered geomagnetically quiet time condition. Thus, our methodology is consistent because the periods more disturbed (for instance, over 1.12) can statistically be considered when Bz is southward and also becomes more disturbed with the increase of the solar wind velocity.

[Figure]

Figure 1. Kp average values in function of IMF's Bz component and solar wind velocity recorded by the OMNI satellite.

Similarly, to Figure 1, we have computed the dependence of Kp to variations of Dst/Sym-H values for the period in this study. Figure 2 shows the Kp average for different ranges of Sym-H (from -68nT to 16nT, steps of 2±1nT). It is clearly seen that with the decrease of Sym-H starting in 0 up to -68nT there is an increase of Kp as well. On the other hand, variations from 0 to 16nT also is noticed an increase of kp up to Kp about 1.3, which is considered quiet condition. In a certain way, the Sym-H from 16 to -16nT nulls each other for periods of quiet condition. This statement is tested in Figure 3.

[Figure]

Figure 2. Dispersion diagram between the kp index in respect to Sym-H values for the year of 2009.

As a matter of comparison, using the same methodology proposed in the manuscript (see Figure 6 of the manuscript text), but considering now only data after 17:00 UT in order to increase the sample space (note in Figure 6 that the behavior of the IL is similar, i.e., the h'IL and ftIL tends to increase/decrease with the increase of kp, respectively) and the same range of Kp, Figure 3 below shows the variability of Sym-H, IMF Bz, solar wind velocity, ΔftIL and Δh'IL in respect to Δkp. It is clearly noticed that with the increase of ΔKp Bz increases to south while Sym-H decreases. This pattern is well defined, evidencing that the magnetic disturbances considered in our study are associated with the direction and intensity of Bz and with the intensity of Sym-H. In this Figure, it is also seen the analysis with solar wind velocity (third panel from the top), delta frequency (fourth panel) and delta height (fifth panel). This presented analysis corroborates with the statement made by the referee and reported by JGR, 113, A05311, 2008, doi:10.1029/2007JA012879, which also corroborates with the statement of question RC1-5 *"An upward movement of an IL would be consistent with a dawn to dusk electric PPEF caused by a southward IMF"*. Summarizing, in comparison of results presented in the manuscript, we can point out that similar results were found and thus validating our study using kp index.

[Figure]

Figure 3. Variability of Sym-H, IMF Bz intensity, solar wind velocity, ΔftIL and Δh'IL in respect to Δkp.

**RC2-6.** Their grammar should be thoroughly improved. They have to send their manuscript to professional grammatical corrections before further review processes.

We carefully read the manuscript and made some corrections in the English grammar. So we hope it now matches the journal standard.

---

## Author Response (AR2)

Summary

I wish to appreciate that the authors have slightly improved the manuscript, whereas the key issues have been unresolved. I had to repeat some off my previous comments, which should be incorporated in their revision.

We thank Referee# 2 for all the comments and suggestions, which by the way improved substantially this report and we acknowledge you for your time on it. Please see below our answers.

Major Comments:

1. The authors have claimed the solar activity in 2009 as something comparable with the Maunder Minimum, citing Zharkova (2020). However, this is probably not true. As I have pointed out, recent studies have proven that the solar activity was much more quiet during the Maunder Minimum than during 2009, on the basis of the cosmogenic isotopes (DOI: 10.1051/0004-6361/201526652; DOI: 10.1051/0004-6361/202140711), the sunspot records (DOI: 10.1093/mnras/stab1155; DOI: 10.3847/1538-4357/abd949), and the visual coronal structures (DOI: 10.1051/swsc/2020035). The authors should cite all these references (not only Carrasco et al., 2021) and keep their implication more conservative.

This part of the text was rewritten (see P8-9, L192-201) and the references suggested were included.

*Using Johann Heinrich Müller's sunspot observations from 1709 (Figure 5 of Hayakawa et al. 2021a), and the sunspot catalog published by the Kislovodsk Mountain Astronomical Station of the Central Astronomical Observatory at Pulkovo for the recent solar cycles (1996–2019), Carrasco et al. (2021) showed that one of the most active years in the Maunder Minimum (1709), was still less active than most years in the Dalton Minimum and also less active than those of the most recent solar minima. Additionally, they mentioned that only the solar activity levels in 2008, 2009, and 2019 were similar to or lower than (in the case of 2008, see Figure 2 of Carrasco et al. 2021) the most probable active day fraction value for 1709. This reinforces how special is of the period chosen here to analyse the possible dependence of ILs on geomagnetic activity. For more detail*

*about Maunder Minimum, see for example Usoskin et al. 2015, 2021; Carrasco et al. 2021; and Hayakawa et al. 2021a,b).*

2. If they wish to state the significance for 2009 in a centennial timescale, their statement on the F10.7 in 2009 does not help the reader to understand the solar variability in the long-term context. This should be replaced to the discussions on sunspot number (after revisions in 2014-2016), citing DOI: 10.1007/s11214-014-0074-2 and DOI: 10.1007/s11207-016-1014-y.

The purpose of this work is to analyze the behavior of ILs to geomagnetic variations during the daytime period. For this, we chose the period 2009 which was considered by the aeronomy community perhaps the quietest of the radio observation era. Following the reviewer suggestions, we have revised several studies about the Sun variability and decided to be superficial in this context, because the scope of this study is not the solar variation, but the analysis of the ILs with respect to geomagnetic variations assuming very little influence or none from the Sun radiation (well-known atmospheric forcing). Thus, we have chosen to be very conservative in the Sun variability discussion, as you can see in this revised version.

3. It is understandable for the authors to value their own studies. Everyone does, indeed. However, the authors have mentioned other authors' studies in the discussion. In this case, they must appropriately address what other authors have achieved in the introduction (even if it is for other geographic sectors).

We understand your point. We have added in the introduction a revision made of several other studies of ILs. Please see P3, L60-74.

*Fujitaka and Tohmatsu (1973) reported that the solar semi-diurnal atmospheric tide can be the dominant cause of the intermediate layers at night and that the vertical drift of the ionizations by the Sq electric field seems to modify the altitude variation of the ILs during this time. Szuszczewicz et al. (1995) found that the ILs are observed throughout the day and in all latitudes that covered the northern and southern hemispheres. Besides that, they also reported the formation of ILs at high altitudes (> 170 km) and a monotonic descent to lower altitudes at rates as high as 8.5 km/h. Rodger et al. (1981) noted that the ILs over South Georgia (54oS, 37oW) are characterized by a prior downward movement of the F-layer, followed by the formation of the intermediate layer and its subsequent drift*

*downwards to about 140 km. They also mentioned that initially this downward movement of the ILs can be at the same rate as the F layer, but decreases as the ILs attained lower altitudes. Mridula and Pant (2021) studied the behavior of ILs over the equatorial location of Thiruvananthapuram and noted that the occurrence of ILs over this sector is higher in the summer and winter solstice and lower in equinoxes. They also showed that the occurrence of this layer is higher in the solar minimum than in the solar maximum period. The possible influence of the gravity waves in determining these characteristics is also discussed by the authors.*

4. I do not think the Kp index is the best tool to visualize the responses of the IL to the overall level of geomagnetic activity. The authors should redo their analyses with the Dst index, which is much more quantitative than the Kp index.

Indeed, the Dst index is much more quantitative than the Kp index and very adequate to case studies due to the better temporal resolution. Our work search for responses of the IL's to geomagnetic activity based on a global scale, and in this case we choose to use the planetary index Kp, largely used by our community and supplied on a daily basis. Also, we are using large temporal samples, that is, one-hour window of IL data considering the previous 9 hours of the geomagnetic condition for about 121 days (one whole season). Therefore, the temporal resolution is not relevant here, as well as the Dst variation in such a range of time. As it was shown in the last revision, we did perform some analysis to answer a question of reviewer #1 that was very similar to the question of reviewer #2 now (see Figure 1 below). It was verified that the increase of ΔKp is compatible with an increase of Bz to south and a decrease of Sym-H, evidencing in this way that the magnetic disturbances considered in our study are associated with the direction and intensity of Bz and with the intensity of Sym-H. So, we do believe that a new analysis with Dst index wouldn't give any different light compared with those got with ΔKp.

[Figure]

Figure 1 – a) Dispersion diagram between the kp index in respect to Sym-H values for the year of 2009. b) Variability of Sym-H, IMF Bz intensity, solar wind velocity, ΔftIL and Δh'IL in respect to Δkp.

Minor Comments

P1L26: ambient ionosphere => ambient ionosphere since 1947 (if the authors insist in using the F10.7 index)

Done.

P2L30 Cite Ken Tapping's works here.

Done.

P8L181: Johann Heinrich Müller's sunspot observations from 1709 => Johann Heinrich Müller's sunspot observations from 1709 (Figure 5 of DOI: 10.3847/1538-4357/abd949)

Done.

P8L183: those of the most recent solar cycles => those of the most recent solar minima

Done.

P8L184: similar to => similar to or lower than.

Done.